# FUSE-Net: Multi-Scale CNN for NIR Band Prediction from RGB Using GNDVI-Guided Green Channel Enhancement

**DOI:** 10.3390/s25134076

**Published:** 2025-06-30

**Authors:** Gwanghyeong Lee, Deepak Ghimire, Donghoon Kim, Sewoon Cho, Byoungjun Kim, Sunghwan Jeong

**Affiliations:** IT Application Research Center, Korea Electronics Technology Institute, Jeonju 54853, Republic of Korea; lightbro@keti.re.kr (G.L.); deepak@keti.re.kr (D.G.); clickmiss123@keti.re.kr (D.K.); swcho@keti.re.kr (S.C.); jun0420@keti.re.kr (B.K.)

**Keywords:** hyperspectral imaging, GNDVI, NIR prediction, multi-scale CNN, MLP-Mixer, *Ocimum basilicum*

## Abstract

Hyperspectral imaging (HSI) is a powerful tool for precision imaging tasks such as vegetation analysis, but its widespread use remains limited due to the high cost of equipment and challenges in data acquisition. To explore a more accessible alternative, we propose a Green Normalized Difference Vegetation Index (GNDVI)-guided green channel adjustment method, termed G-RGB, which enables the estimation of near-infrared (NIR) reflectance from standard RGB image inputs. The G-RGB method enhances the green channel to encode NIR-like information, generating a spectrally enriched representation. Building on this, we introduce FUSE-Net, a novel deep learning model that combines multi-scale convolutional layers and MLP-Mixer-based channel learning to effectively model spatial and spectral dependencies. For evaluation, we constructed a high-resolution RGB-HSI paired dataset by capturing basil leaves under controlled conditions. Through ablation studies and band combination analysis, we assessed the model’s ability to recover spectral information. The experimental results showed that the G-RGB input consistently outperformed unmodified RGB across multiple metrics, including mean squared error (MSE), peak signal-to-noise ratio (PSNR), spectral correlation coefficient (SCC), and structural similarity (SSIM), with the best performance observed when paired with FUSE-Net. While our method does not replace true NIR data, it offers a viable approximation during inference when only RGB images are available, supporting cost-effective analysis in scenarios where HSI systems are inaccessible.

## 1. Introduction

Hyperspectral imaging (HSI) has emerged as a critical technology in remote sensing, gaining increasing attention in diverse fields such as precision agriculture, environmental monitoring, and ecological analysis. HSI captures tens to hundreds of contiguous spectral bands, including near-infrared (NIR) wavelengths, providing rich spectral information for detailed analysis of surface composition and plant physiological characteristics. In particular, the NIR region is closely associated with chlorophyll absorption, making it highly effective for quantitatively assessing vegetation health and growth status. Beyond vegetation monitoring, HSI has been widely applied to a variety of target-specific remote sensing tasks [1,2,3,4,5].

However, hyperspectral imaging systems require expensive equipment and complex operational environments, limiting their practical use in field applications. Challenges such as large device sizes, slow acquisition speeds, large data volumes, and high costs continue to hinder their widespread deployment. In contrast, RGB imaging is affordable and widely accessible but lacks the NIR band, which makes it unsuitable for calculating vegetation indices such as the Normalized Difference Vegetation Index (NDVI) or Green Normalized Difference Vegetation Index (GNDVI). To address this limitation, recent research has increasingly focused on estimating or approximating NIR reflectance from RGB imagery alone [6,7,8].

Previous studies have primarily explored the prediction of NIR bands from RGB images using deep learning frameworks such as U-Net, convolution neural networks (CNN), or generative adversarial networks (GAN). Among these, U-Net, originally developed for biomedical image segmentation, has been widely adopted in various image restoration and transformation tasks, including vegetation area prediction, where its effectiveness has been well demonstrated. More recently, architectures based on MLP-Mixer and Transformers have been introduced to enhance inter-channel representation learning for RGB-to-HSI conversion tasks [9,10,11,12,13,14,15]. However, these approaches generally rely on raw RGB inputs, without explicitly considering the physical significance or functional role of each individual RGB channel. To address this limitation, more proactive utilization of the channel-specific spectral meaning within RGB images has been proposed. For example, the GNDVI, which combines NIR and Green bands, is known to be particularly effective in assessing plant vigor. Compared to the NDVI, the GNDVI responds more sensitively to Green channel information. Therefore, by adjusting or emphasizing the Green channel to mimic NIR-like reflectance characteristics, it is possible to improve the accuracy of the NIR prediction based solely on RGB data.

In this study, we propose a novel RGB transformation method, termed G-RGB, which modifies the Green channel based on the conceptual framework of the GNDVI. The goal was to generate an input representation that structurally resembles the NIR reflectance characteristics using only RGB images. By adjusting the Green channel to approximate the reflectance behavior of the NIR band, the G-RGB transformation enhances the representational capacity of standard RGB imagery, thereby enabling more accurate NIR prediction compared to using the original RGB inputs.

In addition, we introduce FUSE-Net, a novel architecture that integrates the U-Net framework with multi-scale convolutional layers and MLP-based channel learning. The proposed model combines residual blocks for stable feature learning, multi-scale CNNs to capture spatial information at various resolutions, and MLP-Mixer modules to learn inter-channel relationships. Compared to existing models such as U-Net [16], Pix2Pix [9], VGG [17], HFR [18], and self-attention-based networks, FUSE-Net achieves superior prediction performance. In particular, the architecture applies the MLP Channel Mixer twice, before and after the multiscale feature extraction stage, enabling the model to effectively capture not only spatial context, but also spectral interactions across different bands.

## 2. Related Works

### 2.1. NIR Prediction from RGB Imagery

Near-infrared (NIR) imaging has proven invaluable in numerous applications, including vegetation monitoring, material classification, and remote sensing. However, dedicated NIR sensors are often costly or impractical for wide-scale deployment. This has motivated research into estimating NIR reflectance from standard RGB images, enabling broader access to spectral information through learning-based models.

Early approaches attempted hand-crafted mappings between RGB and NIR channels, but such techniques lacked generalizability due to scene dependency. More recent studies have adopted deep learning models, particularly convolutional neural networks (CNNs), to learn data-driven mappings from RGB to NIR [7,19]. These models often rely on supervised learning using paired RGB–NIR datasets, attempting to reconstruct the missing NIR information with minimized spectral error.

Despite the challenging nature of this problem due to the limited spectral resolution of RGB images, researchers have demonstrated that incorporating spatial and contextual cues through deep architectures can significantly enhance NIR prediction accuracy. For instance, NIR synthesis has been used to approximate vegetation indices such as the NDVI and GNDVI [20,21], which require an NIR input that is not directly available from RGB images. This makes NIR prediction a critical pre-processing step for downstream tasks in precision agriculture and environmental monitoring.

### 2.2. Spectral Index Estimation and Vegetation Analysis

Hyperspectral imaging systems leverage a wide range of spectral bands, allowing users to select specific combinations based on application requirements [22]. A prominent application of this flexibility is in the use of vegetation indices, which integrate NIR and visible bands, typically Green or Red, to quantify plant health. These indices have been successfully applied in terrain analysis, crop monitoring, stress detection, and disease identification [19,20,21,23,24].

Among the various vegetation indices, the NDVI and the GNDVI are the most representative. The NDVI employs NIR and Red bands, whereas the GNDVI uses NIR and Green, offering improved sensitivity to chlorophyll content under certain conditions. Since RGB images lack NIR information, directly computing indices like the NDVI or GNDVI is not feasible. To address this, recent research has focused on estimating approximate NIR reflectance from RGB imagery alone. Despite promising results, the limited spectral resolution of RGB still presents a fundamental challenge to accurate NIR reconstruction.

Figure 1 shows a comparison of grayscale, NDVI, GNDVI, and NDWI (Normalized Difference Water Index) representations derived from multispectral data. The NDWI is computed as (Green − NIR)/(Green + NIR) and is commonly used to assess water content in vegetation and to monitor drought stress and irrigation effectiveness.

### 2.3. Deep Learning Architectures for Spectral Reconstruction

Various network architectures have been proposed to support RGB-to-NIR or RGB-to-hyperspectral image conversion tasks. Among them, encoder–decoder structures such as U-Net have shown notable success.

#### 2.3.1. U-Net for Spectral Prediction

The U-Net architecture, originally proposed by Ronneberger et al. [16] for bio-medical image segmentation, is a deep learning model based on an encoder–decoder framework. In this architecture, the encoder progressively learns abstract feature representations, while the decoder gradually reconstructs the original resolution. Crucially, skip connections are employed to link intermediate encoder features directly to the corresponding decoder layers, thereby preserving fine-grained spatial details.

In the context of hyperspectral image restoration, prediction, and classification, U-Net has proven to be highly effective, as it can simultaneously leverage both spatial and spectral information. For example, Dixit et al. [25] used a 3D U-Net with three-dimensional convolutional layers to effectively denoise hyperspectral images. Similarly, Shukla et al. [7] introduced a residual U-Net structure with attention blocks to predict NIR images from RGB inputs. These extended versions of U-Net have demonstrated promising results in RGB-to-HSI conversion tasks and have become foundational to various hybrid network architectures.

#### 2.3.2. MLP-Mixer for Band-Wise Feature Modeling

In the field of deep-learning-based computer vision, architectures beyond CNNs and Transformers have been proposed, including the MLP-Mixer framework [15]. The MLP-Mixer is a novel architecture that completely separates spatial and channel dimensions, learning information along each axis independently. While conventional CNNs rely on kernel-based local feature extraction, and Transformers utilize self-attention mechanisms to model long-range dependencies, the MLP-Mixer instead employs repeated application of simple dense (fully connected) layers, allowing for comprehensive integration of information across the entire input.

The MLP-Mixer extracts features through two sequential stages: spatial mixing and channel mixing. First, the input is divided into non-overlapping patches, and spatial mixing captures relationships among these patches. Subsequently, channel mixing models interactions between the feature channels. This architecture enables a structurally simple yet powerful network that can efficiently capture global semantic information across the entire image. Moreover, its independent processing of channel dimensions makes it particularly well suited for data with rich spectral information such as hyperspectral imagery, where learning inter-band correlations is critical.

However, one limitation of the MLP-Mixer is its relatively weak ability to directly capture spatial positional information compared to convolutional or attention-based mechanisms. To address this drawback and ensure stable learning, it is common to incorporate techniques such as residual connections and layer normalization into the architecture.

#### 2.3.3. Multi-Scale Networks for RGB-to-NIR Translation

Multi-scale networks are widely used across various computer vision tasks, as they integrate features extracted at different spatial resolutions to simultaneously capture global context and local details [26,27,28]. Low-resolution inputs are advantageous for recognizing broad semantic structures, whereas high-resolution inputs are effective for detecting fine boundaries and detailed shapes. This architecture improves robustness to variations in object size and enhances predictive performance. However, it also tends to increase model complexity, requiring a careful balance between computational efficiency and performance. Lim et al. [29] significantly improved the results of single-image super-resolution tasks using a residual-based multi-scale network. Similarly, Tao et al. [30] proposed a scale-recurrent structure to effectively handle complex long-range dependencies for image deblurring. In the hyperspectral imaging domain, Zhang et al. [31] introduced a Multi-Scale Dense Network that simultaneously leverages multi-scale information, leading to substantial improvements in hyperspectral image classification performance.

## 3. GNDVI-Guided RGB (G-RGB)

Vegetation indices such as the NDVI, GNDVI, and NDWI are widely used for assessing plant health and vigor due to their simplicity and interpretability. However, these indices fundamentally rely on near-infrared (NIR) reflectance values, typically obtained from hyperspectral or multispectral sensors, for accurate computation. Since standard RGB images lack NIR bands, the direct calculation of such indices is infeasible with RGB-only inputs. This dependency limits their applicability in low-cost imaging systems, where acquiring NIR information is not feasible.

In conventional hyperspectral image processing, vegetation indices such as the GNDVI are calculated using the NIR and Green bands to analyze plant vigor and reflectance characteristics. This relationship is mathematically expressed in Equation (Equation 1).(1)$$\mathrm{GNDVI}=\frac{\mathrm{NIR}-\mathrm{Green}}{\mathrm{NIR}+\mathrm{Green}}$$

However, since standard RGB images do not contain NIR band information, this study proposes a modified approach inspired by the concept of the GNDVI. Specifically, we introduce a GNDVI-guided correction-based RGB transformation method, referred to as G-RGB, which enables the approximation of NIR reflectance characteristics from RGB inputs. The transformation is learned based on an initial paired RGB–NIR training dataset.

Unlike conventional RGB–NIR learning-based methods that learn a direct mapping from radiometrically calibrated and aligned RGB composite images to the NIR band using paired datasets, the proposed G-RGB approach introduces an additional preprocessing step inspired by the GNDVI. Specifically, instead of using RGB images as-is, we first apply a GNDVI-guided transformation that adjusts the green channel to form a spectrally enriched G-RGB representation. This transformation is learned during training using paired RGB–NIR data and subsequently applied to standard RGB images at inference. The key distinction lies in the input structure and learning objective: while conventional methods rely on RGB images derived from multispectral cameras with radiometric calibration and spatial alignment, and learn RGB→NIR mappings directly [8], our approach transforms standard RGB images into a domain that better approximates NIR characteristics before prediction. This design improves generalization when working with uncalibrated RGB images at inference time and is particularly well suited for low-cost systems where NIR sensors are only accessible during an initial calibration phase. Equations (Equation 2) and (Equation 3) describe the linear combination used in the G-RGB transformation, which is designed to enhance NIR prediction accuracy. In this equation, λ∈[0,1] represents an adjustment coefficient, empirically set to the optimal value of 0.5 based on experimental validation.

The use of a linear combination of *R* and *G* channels to approximate NIR is motivated by the spectral proximity of these bands to the NIR region. The *R* band (approximately 600–700 nm) lies near the red-edge transition zone, where reflectance characteristics strongly correlate with those in the lower NIR range. Similarly, the *G* band (approximately 500–600 nm) captures vegetation structural information and shows moderate correlation with NIR reflectance in plant canopies. In contrast, the *B* band generally exhibits weak correlation with NIR and is more susceptible to atmospheric and noise effects; thus, it is excluded from the combination. A linear model was adopted due to its simplicity and effectiveness, and λ in Equation (Equation 3) controls the relative contribution of the *R* and *G* bands. This approach provides a practical and interpretable baseline for pseudo-NIR reconstruction.

Let *R*, *G*, and *B* represent the red, green, and blue channel intensities, respectively. The transformed Green channel $G^{\prime }$ is computed as follows: (2)$$\hat{\mathrm{NIR}}=G+\left(G-R\right)$$(3)$$G^{\prime }=G+\lambda \left(\hat{\mathrm{NIR}}-G\right)$$

The G-RGB image is generated from the original RGB input through the following steps:Channel Separation: The Red (*R*), Green (*G*), and Blue (*B*) channels are extracted from the original RGB image.Approximate NIR Calculation: An approximate NIR reflectance value $\hat{\mathrm{NIR}}$ is estimated using the difference between the Green and Red channels.Green Channel Adjustment: A new corrected Green channel $G^{\prime }$ is computed based on the estimated NIR and the original Green value, modulated by the adjustment coefficient λ.G-RGB Construction: The final G-RGB image is constructed by combining the adjusted Green channel ($G^{\prime }$) with the original Red and Blue channels.

In cases where the original RGB image contains an overly bright Green channel or an unusually dark Red channel, the computed corrected Green value $G^{\prime }$ may exceed the valid range of [0, 1], leading to saturation issues. This can result in the loss of meaningful information. To mitigate this, a clamping technique is employed during the G-RGB generation process. This approach limits excessive adjustments to the Green channel and avoids unnecessary corrections, thereby preserving image fidelity. Specifically, the adjustment range is constrained using a simple conditional function, as defined in Equation (Equation 4), to ensure that the corrected channel values remain within a valid range and do not exceed the standard 8-bit channel maximum of 255.(4)$$G^{\prime }=\left\{\begin{array}{cc} 0, & \mathrm{if}\;G+\lambda (\hat{\mathrm{NIR}}-G)<0\\ G+\lambda (\hat{\mathrm{NIR}}-G), & \mathrm{if}\;0\leq G+\lambda (\hat{\mathrm{NIR}}-G)\leq 1\\ 1, & \mathrm{if}\;G+\lambda (\hat{\mathrm{NIR}}-G)>1 \end{array}\right.$$

To analyze the differences between original RGB images and those transformed using the proposed G-RGB method, we conducted both visual and quantitative comparisons using representative sample images from well-known datasets, including the COCO-Dataset [32], ImageNet [33], Plant Village [34], and GoPro-Dataset [35]. Specifically, we examined the changes in Green channel values and overall image appearance.

First, the L1 distance (Mean Absolute Error, MAE) was used to intuitively evaluate the pixel-wise differences between the original and transformed images. In addition, L2 distance (Root Mean Square Error, RMSE) was employed to emphasize and detect regions with large discrepancies, effectively suppressing minor pixel-level noise [36]. To assess broader color and brightness variations in the Green channel, we also used Histogram Similarity (HS), which provides a global comparison of pixel value distributions rather than individual pixel differences.

Table 1 presents the quantitative evaluation results comparing the original RGB images and those transformed using the proposed G-RGB method. The L1 and L2 analyses indicate that, although the G-RGB images showed noticeable differences across all four datasets, the corresponding L2/L1 ratios (1.90, 2.08, 1.99, and 2.02) remained below the commonly accepted threshold of 2.5. Even accounting for the general tendency of L2 to exceed L1 due to its sensitivity to larger errors, the absence of excessive increases suggests that no significant distortions occurred in localized regions. The Histogram Similarity (HS) results remained consistently high across all datasets, with values above 0.97, indicating that the G-RGB method preserved most of the original color characteristics. Furthermore, when HS was calculated specifically for the Green channel (HS(G)), the values were 0.7820 (COCO), 0.9809 (ImageNet), 0.9241 (Plant Diseases), and 0.9603 (GoPro). These results suggest that although the Green channel was more strongly adjusted in the COCO dataset, the G-RGB transformation was able to minimize unnecessary color distortion in the other datasets, while maintaining a balanced correction effect.

Figure 2 qualitatively demonstrates the effectiveness of the proposed G-RGB correction method on real-world images. Compared to the original RGB inputs, the transformed images exhibit an enhanced representation of green areas, which is particularly evident in vegetation-rich scenes. These enhancements suggest a promising potential for improved NIR band prediction performance using the G-RGB representation.

## 4. Fusion-Based Spectral Estimation Network (FUSE-Net)

Figure 3 shows the overall architecture of the proposed FUSE-Net, designed for the prediction of a specific NIR band. The network adopts an encoder–decoder framework based on the U-Net architecture, enabling the effective extraction of both spatial and spectral features from the input G-RGB image. These features are then used to reconstruct the corresponding NIR band image. The architecture incorporates multi-scale feature learning and channel-wise transformations to enhance prediction accuracy.

The FUSE-Net processes information through the following sequence of operations. The input image has a shape of (H,W,3), and it is first passed through a convolutional block composed of Conv–BatchNorm–ReLU operations, which expands the channel dimensions to 64. At this stage, a Convolutional Block Attention Module (CBAM) [37] is integrated into the convolutional block. This module is designed to simultaneously learn spatial and channel-wise attention, allowing the network to assign greater weights to more informative features.

As illustrated in Figure 3, the CBAM consists of two sequential sub-modules: a Channel Attention Module (CAM) and a Spatial Attention Module (SAM). The CAM refines channel-wise feature responses using global average pooling and global max pooling, followed by two fully connected layers with ReLU and Sigmoid activations to generate channel attention weights, which are then applied to the input feature map via element-wise multiplication. Subsequently, the SAM applies spatial attention by first performing average pooling and max pooling across the channel dimension, followed by a 7 × 7 convolution to produce a spatial attention map. This spatial attention map is likewise multiplied with the feature map to emphasize informative spatial regions. By integrating the CBAM at multiple stages of the FUSE-Net architecture, including after each encoder block, within the bottleneck, and in the decoder upsampling blocks, our network adaptively enhances both spectral and spatial representations, thereby improving the accuracy and robustness of NIR estimation.

This attention-based feature extraction process is recursively applied at each stage of the encoder. As the image passes through the encoder, the spatial resolution is halved at each stage via max-pooling, while the number of channels is gradually increased to 64, 128, 256, and finally 512.

The final output of the encoder is forwarded to the bottleneck module, which serves as the core of the proposed FUSE-Net architecture. This module comprises three main components. First, an MLP Channel Mixer block is applied to the input features to model global channel-wise relationships using dense layers along the channel axis. This component enables the network to effectively capture spectral interactions and latent dependencies among bands. Second, to incorporate features at multiple spatial scales, a Multi-Scale CNN module is introduced. This block consists of three parallel convolutional operations with different kernel sizes (3 × 3, 5 × 5, and 7 × 7), allowing the extraction of visual features at varying resolutions. The resulting outputs are concatenated along the channel dimension and merged into a unified tensor with a shape of (H/16,W/16,N). Third, another MLP Channel Mixer block is applied to further refine the channel interactions following multi-scale processing. A residual skip connection is also established between the input and the final output of this bottleneck block to prevent feature degradation. The output of the bottleneck is then passed through a convolutional block with 1024 filters and an integrated CBAM module, refining the high-dimensional aggregated representation.

The decoder is constructed in the reverse direction to the encoder, progressively restoring spatial resolution at each stage through upsampling operations. At each level, the upsampled feature maps are concatenated with the corresponding encoder outputs via skip connections, allowing the network to recover lost spatial information.

Each upsampling block is composed of a Conv–BN–ReLU structure with an embedded CBAM module, consistent with the design used in the encoder. The number of channels is reduced in a stepwise manner: 512 → 256 → 128 → 64. This structure effectively fuses the multi-resolution, attention-enhanced features extracted in the encoder and contributes to the accurate reconstruction of the predicted output image.

### 4.1. Bottleneck Structure for Integrated Spectral–Spatial Learning

The core of FUSE-Net lies in the bottleneck block located between the encoder and decoder. This region serves as the primary site for information compression and transformation within the network. Rather than using a conventional convolutional block, the bottleneck is designed as a fusion of three specialized modules arranged sequentially: (1) an MLP Channel Mixer, (2) a Multi-Scale CNN, and (3) a second MLP Channel Mixer. This structure enables the model to jointly learn global channel-wise dependencies, multi-scale spatial features, and their integrated representations.

#### 4.1.1. MLP Channel Mixer

The adoption of the MLP-Mixer architecture is theoretically motivated by its ability to model long-range dependencies across spectral channels, which is essential for accurate spectral reconstruction. While CNNs are effective for spatial feature extraction, they often lack the ability to capture global spectral relationships. By applying fully connected layers along the channel axis, the MLP Channel Mixer captures complex inter-channel interactions that are critical for inferring latent NIR structures from G-RGB inputs. This design provides a computationally efficient alternative to attention-based mechanisms, while still enabling robust spectral correlation modeling required in hyperspectral estimation tasks.

In the first stage, an MLP Channel Mixer block is applied to model relationships along the channel axis. Inspired by the MLP-Mixer architecture originally derived from Transformer networks, this variant is customized for channel-wise operations. In this study, the MLP Channel Mixer is constructed using batch normalization, dense layers, and residual connections.

Let the input feature map be denoted as $X\in R^{H\times W\times C}$, where *H* and *W* are the spatial dimensions and *C* is the number of channels. Given an input feature map *X*, the data first undergo batch normalization, followed by flattening and dense transformations. The detailed processing steps are described in Equation (Equation 5) and further explained below. This MLP Channel Mixer is applied twice within the bottleneck: once at the beginning and again at the end. To provide greater modeling capacity and enable richer inter-channel interactions, the first dense layer (Dense1) expands the channel dimension by a factor of 4 (i.e., from *C* to 4C), introducing nonlinearity and allowing the network to learn more expressive feature representations. The second dense layer (Dense2) then projects the dimension back to *C*, restoring compatibility with the original input. Furthermore, a residual skip connection is incorporated between the input and the output of the MLP Channel Mixer block. This residual connection helps mitigate the vanishing gradient problem, facilitates stable optimization, and preserves important features from the original input throughout the learning process.
(5)$$\begin{array}{cc} X_{\mathrm{bn}} & =\mathrm{BatchNorm}(X),\\ X_{r} & =\mathrm{Reshape}\left(X_{\mathrm{bn}}\right)\in R^{(H\cdot W)\times C},\\ Z & ={\mathrm{Dense}}_{2}\left({\mathrm{Dense}}_{1}\left(X_{r}\right)\right),\\ Z_{\mathrm{res}} & =Z+X_{r},\\ Y & =\mathrm{Reshape}\left(Z_{\mathrm{res}}\right)\in R^{H\times W\times C} \end{array}$$

#### 4.1.2. Multi-Scale CNN (MSC) Block

In the second stage, a Multi-Scale CNN (MSC) block is introduced to capture visual features at various spatial resolutions. This module consists of three parallel convolutional layers with different kernel sizes (3 × 3, 5 × 5, and 7 × 7) designed to extract multi-scale spatial information. A subsequent 1 × 1 convolution is applied to adjust the channel dimension and refine the concatenated features. To further enhance the representation, a second MLP Channel Mixer block is applied to the output of the MSC, mirroring the structure described in Section 4.1.1. This enables additional modeling of inter-channel dependencies, complementing the multi-scale feature extraction. The corresponding operations are defined in Equation (Equation 6).(6)$$\begin{array}{cc} F_{1} & ={\mathrm{Conv}}_{3\times 3}\left(F\right),\\ F_{2} & ={\mathrm{Conv}}_{5\times 5}\left(F\right),\\ F_{3} & ={\mathrm{Conv}}_{7\times 7}\left(F\right),\\ F_{\mathrm{concat}} & =\mathrm{Concat}\left(F_{1},F_{2},F_{3}\right)\in R^{H\times W\times C},\\ F_{\mathrm{msc}} & ={\mathrm{Conv}}_{1\times 1}\left(F_{\mathrm{concat}}\right),\\ F_{\mathrm{final}} & =\mathrm{MLP}\_\mathrm{Channel}\_\mathrm{Mixer}\left(F_{\mathrm{msc}}\right)+F_{\mathrm{msc}}, \end{array}$$

## 5. Dataset Construction

In hyperspectral image research, publicly available datasets such as Indian Pines, Pavia University, and Houston2013 have been widely used [38,39,40]. These datasets are primarily designed for hyperspectral image classification tasks and typically consist of single hyperspectral images containing tens to hundreds of spectral bands with pixel-wise class labels. However, these datasets are not suitable for spectral prediction tasks, such as predicting specific NIR bands from RGB images, because they do not contain precisely aligned RGB–HSI image pairs. For such prediction tasks, it is essential to have datasets that include RGB and hyperspectral images of the same scene, captured from the same position, at the same time, and under consistent lighting conditions.

Constructing such datasets requires an imaging setup capable of simultaneously acquiring RGB and hyperspectral data. Due to the large file sizes of hyperspectral images (approximately 2 GB per image), the high cost of equipment, and operational constraints, it is extremely rare to find publicly available datasets that reflect real-world conditions and provide high-resolution spatially aligned RGB-NIR data.

Accordingly, to experimentally validate the performance of the proposed G-RGB–based NIR prediction model, we constructed a custom dataset tailored to the needs of the study. The data were collected inside a greenhouse located in Gimje, Jeollabuk-do, South Korea. The target object for image acquisition was basil (*Ocimum basilicum*) leaves.

The data acquisition process was conducted under natural lighting conditions, with additional control provided by two Fomex H1000 lights positioned on either side of the scene. Both the RGB and HSI cameras were mounted on tripods and carefully aligned at a fixed distance and angle to ensure consistent image capture of the same scene at the same moment. The hyperspectral images were captured using an HERA-VIS camera, which covers the 400–1000 nm spectral range across 120 bands. RGB images were acquired using an ELP-4K-USB camera. A total of 50 RGB–HSI image pairs were collected. The dataset was randomly split into 80% for training and 20% for validation. Although the number of image pairs was limited, each hyperspectral image contained dense spatial and spectral information, with 120 bands per pixel, yielding a substantially rich dataset. Representative sample images from both modalities are shown in Figure 4.

The acquired hyperspectral images, originally captured at a resolution of 512 × 640 × 110, were normalized based on the maximum reflectance value and resized to 256 × 256 × 110. The preprocessed data were saved in NumPy array format, for efficient handling during training.

Based on this dataset, we conducted training, validation, and performance evaluation of the proposed G-RGB–based NIR band reconstruction model.

## 6. Experimental Results

In this section, we present the experimental results obtained by evaluating the proposed G-RGB input and FUSE-Net architecture. The experiments were conducted in two stages: (1) selecting the optimal NIR band combination based on Principal Component Analysis (PCA), and (2) benchmarking the performance of FUSE-Net and existing models using both the original RGB and G-RGB inputs. All experiments were performed using a custom-collected RGB–HSI paired dataset acquired under controlled conditions.

### 6.1. Evaluation Metrics

To assess the performance of NIR band prediction from RGB or G-RGB inputs, we used four quantitative metrics that jointly captured both spatial fidelity and spectral accuracy. These included Mean Squared Error (MSE), Peak Signal-to-Noise Ratio (PSNR), Structural Similarity Index (SSIM), and Spectral Correlation Coefficient (SCC). For each metric, the average value across the test dataset was reported as the final performance score. SSIM, originally proposed by Wang et al. [41], is a widely adopted metric in image reconstruction tasks, due to its ability to reflect structural fidelity. It has also been extensively used in recent restoration-based vision studies [42,43]. In addition, SCC was used to evaluate the spectral consistency between the predicted and ground-truth reflectance profiles. Based on the Pearson correlation coefficient, SCC quantifies the linear relationship between spectral vectors, offering a complementary perspective to distortion-based metrics. By focusing on spectral correlation, SCC provides further insight into how well the predicted outputs preserve the original spectral characteristics [44].

Mean Squared Error (MSE): Measures the average squared difference between the predicted NIR image $\hat{I}$ and the ground truth *I*. Lower values indicate better prediction accuracy.(7)$$\mathrm{MSE}=\frac{1}{mn}\sum_{i=1}^{m}\sum_{j=1}^{n}{\left(I\left(i,j\right)-\hat{I}\left(i,j\right)\right)}^{2}$$Peak Signal-to-Noise Ratio (PSNR): Represents the ratio between the maximum possible pixel value $MAX_{I}$ and the mean squared error, expressed in decibels. Higher values indicate better image quality.(8)$$\mathrm{PSNR}=10\cdot \log_{10}\left(\frac{MAX_{I}^{2}}{\mathrm{MSE}}\right)$$Spectral Correlation Coefficient (SCC): Evaluates the correlation between predicted and ground truth spectral values across channels. A higher SCC value (close to 1) indicates a stronger linear relationship and better spectral preservation in the reconstructed NIR output.
(9)$$\mathrm{SCC}=\frac{\sum_{i=1}^{N}\left(y_{i}-\bar{y}\right)\left({\hat{y}}_{i}-\bar{\hat{y}}\right)}{\sqrt{\sum_{i=1}^{N}{\left(y_{i}-\bar{y}\right)}^{2}}\sqrt{\sum_{i=1}^{N}{\left({\hat{y}}_{i}-\bar{\hat{y}}\right)}^{2}}}$$Structural Similarity Index Measure (SSIM): Compares two images *x* and *y* in terms of luminance (μ), contrast (σ), and structure ($\sigma_{xy}$). Values closer to 1 imply higher perceptual similarity.(10)$$\mathrm{SSIM}\left(x,y\right)=\frac{\left(2\mu_{x}\mu_{y}+C_{1}\right)\left(2\sigma_{xy}+C_{2}\right)}{\left(\mu_{x}^{2}+\mu_{y}^{2}+C_{1}\right)\left(\sigma_{x}^{2}+\sigma_{y}^{2}+C_{2}\right)}$$

### 6.2. Training Setup and Hyperparameters

All experiments were conducted in a system equipped with an NVIDIA GeForce RTX 4090 GPU (NVIDIA Corporation, Santa Clara, CA, USA), 512 GB RAM (Samsung Electronics, Suwon, Republic of Korea), and an AMD Ryzen Threadripper PRO 3955WX 16-core CPU (Advanced Micro Devices, Santa Clara, CA, USA). The implementation was carried out in Python 3.8 using the TensorFlow framework.

Each model was trained for up to 450 epochs. The learning rate was reduced if the validation loss did not improve for 10 consecutive epochs. Training was stopped early if no improvement was observed over 30 consecutive epochs. Input image sizes were adjusted based on the model requirements: 64 × 64 × 3 for the Transformer model, 128 × 128 × 3 for Pix2Pix, and 256 × 256 × 3 for the remaining models. Aside from the input size, all other preprocessing steps were kept consistent. Table 2 summarizes the training hyperparameters used for all models, except for input size, which was adjusted depending on the architecture.

### 6.3. Selection of Optimal NIR Band

In the task of predicting NIR bands from RGB images, the choice of which NIR bands or combinations to target has a significant impact on model performance. Given the large number of channels within the NIR region, identifying an optimal subset of bands is particularly challenging.

In previous hyperspectral classification or reconstruction studies, it was common to either use the full spectral range or select continuous ranges of bands [45]. However, the objective of this study was to reconstruct a small number of meaningful NIR bands using only limited RGB information. Therefore, an ablation study was conducted to determine the optimal combination of target NIR bands for prediction.

The acquired hyperspectral dataset contained a total of 120 spectral bands. For the purpose of NIR analysis, we focused on bands indexed from 70 to 120, which correspond to wavelengths in the 750–1000 nm range, lying beyond the spectral sensitivity of standard RGB cameras.

To identify the most informative spectral bands within the NIR region, we applied Principal Component Analysis (PCA), a dimensionality reduction technique that transformed the original correlated spectral bands into a set of orthogonal principal components, each representing a direction of maximum variance in the data.

The input matrix for PCA was structured such that each row represented the spectral reflectance vector of a single pixel, and each column corresponded to one NIR band (bands 70–119). PCA was computed based on a covariance matrix of the input data using scikit-learn’s PCA implementation. To evaluate the importance of each spectral band, we calculated cumulative loading scores by summing the absolute values of loadings from the first three principal components (PC1–PC3). The top 20 bands with the highest cumulative loading scores were selected as the most informative bands for NIR estimation. The PCA results are illustrated in Figure 5, where 5a and 5b show 2D and 3D projections of pixel-wise spectral reflectance vectors, and 5c presents the cumulative PCA loading scores of the individual spectral bands used for band selection.

To ensure that the selected NIR bands captured a diverse range of physiologically relevant information, we divided the NIR region (750–1000 nm) into three subgroups based on the known physiological and optical characteristics of plant spectral reflectance. Specifically, the Early NIR range (700–800 nm) corresponds to the region where chlorophyll absorption gradually decreases and reflectance rapidly increases. The Mid NIR range (800–900 nm) is characterized by strong reflectance variations driven by plant cellular structure. The Late NIR range (900–1000 nm) is sensitive to water absorption, providing information related to plant water status. This grouping strategy was not a simple equal split, but was theoretically designed to align with the known spectral properties of vegetation in the NIR region. By selecting representative bands from each subgroup, the final band combinations could better represent the full physiological variability within the NIR spectrum.

Figure 6 shows the average normalized reflectance spectra of all HSI samples across the 400–1000 nm wavelength range. Individual green curves represent sample-wise spectra, while the bold black line denotes the overall mean reflectance. The overlaid red, yellow, and orange boxes delineate the Early NIR (70–80), Mid NIR (81–95), and Late NIR (96–120) band groups, respectively.

Group 1: Early NIR (70-80 Band Index)Group 2: Mid NIR (81-95 Band Index)Group 3: Late NIR (>95 Band Index)

Each band group possesses distinct spectral characteristics, which may have varying effects on a model’s learning dynamics and prediction accuracy. Accordingly, for each group, one band was randomly selected to form a three-band combination, resulting in a total of 20 different NIR band combinations (e.g., (70, 93, 99), (83, 97, 108), etc.).

All experiments were conducted under identical dataset and training configurations. A standard U-Net model was used as the baseline, with G-RGB images as input and the selected three NIR bands as the output. For each combination, the performance was evaluated using 50 samples, and commonly used image quality metrics were computed. Figure 7 presents the performance graph for 10 selected NIR band combinations, while Table 3 summarizes the top six performing combinations out of all 20 evaluated sets. Among them, the combination of (70, 91, 108) demonstrated the best performance across all evaluation metrics, consistently surpassing the overall average scores. This particular band set was found to effectively capture the spectral diversity of the NIR region, as its indices were well distributed across the informative wavelengths. As a result, this combination was used as the default configuration for the subsequent experiments.

### 6.4. Benchmarking G-RGB and the Proposed Model Against Conventional Approaches

This section presents a comprehensive evaluation of the proposed G-RGB input representation and the FUSE-Net architecture. The primary objective was to verify whether the G-RGB images, which were generated using histogram alignment and enhancement techniques based on the Green channel, provided better performance in NIR band prediction compared to the original RGB images. Several models were evaluated, including U-Net [16], Transformer-based models [46], Pix2Pix [17], HFR [18], VGG-based architectures [47], attention-enhanced U-Net models previously used for NIR prediction [7], and the proposed FUSE-Net. Each model was trained and tested using both original RGB and G-RGB images as input to ensure a fair comparison.

The models were designed to predict three selected NIR bands (band indices: 70, 91, and 108) using RGB–HSI paired datasets. Table 4 summarizes the average performance of each model with both original RGB and G-RGB inputs using MSE, PSNR, SCC, and SSIM metrics. The combination of the proposed FUSE-Net model and G-RGB input achieved the best performance, showing the highest PSNR, SCC, and SSIM values among all the compared methods.

To evaluate the generalization performance of the proposed model and verify the consistency of the input strategy, we conducted an additional 5-fold cross-validation using the same dataset. The dataset of 50 paired RGB–HSI samples was randomly partitioned into five folds, each containing approximately 10 samples. The splitting was performed using a stratified sampling approach to ensure a balanced distribution of plant conditions across folds. Each model was independently trained and tested with both the original RGB and G-RGB inputs, using identical fold partitions. The same NIR band combination (bands 70, 91, 108) was consistently used across all folds to maintain comparability and fairness in the evaluation. For each fold, we computed the mean and standard deviation of the evaluation metrics. The results confirmed that the G-RGB input consistently yielded a better and more stable prediction performance, not only in single-split experiments but also across the full cross-validation process. In particular, the proposed FUSE-Net model achieved the lowest Mean Squared Error (MSE) and the highest Peak Signal-to-Noise Ratio (PSNR), Spectral Correlation Coefficient (SCC), and Structural Similarity Index Measure (SSIM) scores across all folds. The final mean and standard deviation values are reported in Table 5, providing a comprehensive assessment of the model’s stability and generalization under varying data partitions. These results indicate that the model maintained strong generalization capabilities regardless of data partitioning.

Figure 8 provides visual comparisons of the NIR band prediction results generated by each model, using both the original RGB and G-RGB images as input. The results for the same test samples are shown side by side. Across most models, the predictions derived from G-RGB inputs exhibit visibly improved clarity, particularly in the delineation of basil leaf regions. These outputs display sharper edges, stronger contrast, and more coherent spatial structure when compared to those generated from the original RGB inputs. In many cases, the boundaries between the foreground (basil leaves) and background are more distinctly defined, indicating that the G-RGB transformation enhanced the model’s ability to isolate and reconstruct vegetative structures.

A closer examination of selected models with high quantitative scores is shown in Figure 9. The U-Net and VGG-based models tended to produce relatively blurry predictions, with a noticeable loss of fine structural detail, especially around the edges of the basil leaves. The U-Net with attention showed improved separation of background areas but struggled to accurately reproduce the shape and intensity gradients within the basil regions. This suggests that while attention mechanisms help with general localization, they may not be sufficient for capturing intricate vegetative structures under limited spectral input.

In contrast, the proposed FUSE-Net model demonstrated the most faithful reconstruction to the ground truth images. Its predictions maintain clear object boundaries and preserve fine-grained textural features. Notably, FUSE-Net’s outputs show less noise in the background and more accurate intensity gradients across the basil area, reflecting both improved spatial coherence and spectral sensitivity. These results highlight the effectiveness of combining the G-RGB input with multi-scale convolutional and channel-wise feature modeling for NIR band restoration tasks.

### 6.5. Ablation Studies

#### 6.5.1. Impact of FUSE-Net Modules on Performance

To assess the contribution of the individual architectural components in FUSE-Net, we conducted an ablation study by selectively removing the channel-wise MLP Mixer block and the multi-scale CNN fusion module. Four model variants were evaluated: (1) the full FUSE-Net, (2) a model without the MLP Mixer block (Variant B), (3) a model without the multi-scale CNN module (Variant C), and (4) a model without both components (Variant A).

To isolate the effect of architectural modifications, all experiments were conducted using the same standard RGB input. This setup ensured that the performance differences could be attributed solely to the inclusion or exclusion of each architectural block.

Table 6 presents the performance of all model configurations in terms of MSE, PSNR, SSIM, and SCC. The full FUSE-Net model achieved the highest overall performance, with particularly strong results in SSIM (0.9406) and SCC (0.9887), reflecting superior structural and spectral fidelity. In contrast, removing both components (Variant A) led to a notable decline in performance, highlighting the critical role of both the channel-wise MLP Mixer and the multi-scale CNN module in prediction quality.

#### 6.5.2. Effect of the λ Parameter on Model Performance

To determine the optimal value of the adjustment coefficient λ in Equation (Equation 3), we conducted a sensitivity analysis by varying λ from 0.1 to 1.0 in increments of 0.1. For each setting, a G-RGB image was generated and used as input to the FUSE-Net model to predict the corresponding NIR band images. Figure 10 presents qualitative examples of G-RGB images produced with different λ values, illustrating how varying the correction strength affected the visual appearance. Table 7 shows the corresponding MSE, PSNR, and SSIM scores for each λ. Based on these results, λ=0.5 was selected as the default value, as it yielded the best overall trade-off between visual fidelity and quantitative accuracy.

#### 6.5.3. Functional Evaluation of Predicted NIR for Vegetation Analysis

To evaluate the practical utility of the predicted NIR bands, we conducted a functional analysis using the GNDVI, which is commonly used to monitor plant health, water stress, and vegetation dynamics. We generated GNDVI maps using predicted NIR from two input variants: (1) RGB images and (2) G-RGB transformed images. These GNDVI maps were then compared against ground truth GNDVI maps computed using the true NIR from hyperspectral data.

The evaluation was performed on all 50 samples. For each sample, we computed three similarity metrics between the predicted GNDVI and the ground-truth GNDVI Root Mean Squared Error (RMSE), Pearson correlation coefficient, and histogram similarity.

As shown in Table 8 and Figure 11, the GNDVI maps derived from G-RGB-based NIR predictions achieved notably lower RMSE and higher histogram similarity compared to those derived from RGB-based predictions, indicating closer visual and distributional alignment with the ground truth. Specifically, compared to RGB-based pseudo-NIR, the G-RGB-based approach reduced the RMSE by 1.71×, slightly decreased the Pearson correlation to 0.98×, and increased the histogram similarity by 4.54×. Although the RGB-based method yielded a marginally higher Pearson correlation, the overall results suggest that G-RGB-based pseudo-NIR provides more reliable support for vegetation analysis when true NIR is unavailable.

## 7. Conclusions and Discussion

In this study, we proposed a multi-scale CNN-based NIR prediction framework incorporating Green channel correction derived from the concept of the GNDVI. The proposed G-RGB method effectively emphasized vegetation-related reflectance characteristics, while preserving the structural information of the original RGB image. This adjustment significantly improved the model’s ability to learn NIR spectral features.

In addition, the FUSE-Net architecture, which integrates residual blocks, multi-scale convolutional layers, and MLP-based spectral learning within an encoder–decoder framework, successfully captured spatial–spectral correlations and demonstrated superior performance compared to conventional models.

Quantitative evaluations based on MSE, PSNR, and SSIM consistently showed that the proposed model outperformed baseline methods. Notably, the use of the G-RGB input led to a clear performance improvement, suggesting that the adjusted Green channel contributed positively to NIR spectrum prediction. Furthermore, a functional comparison using the GNDVI confirmed that pseudo-NIR predicted from G-RGB inputs better approximated vegetation index behavior than RGB-based predictions. These results also highlight the broader potential of combining image processing techniques and deep learning for hyperspectral applications.

Despite the strong performance, several limitations remain. The training and validation datasets used in this study were acquired under controlled lighting conditions and focused solely on basil samples. Therefore, the generalizability of the model to broader conditions and other plant types may be limited. Future work should involve collecting more diverse datasets under varying conditions to improve the model’s robustness and extend its applicability across real-world scenarios.

Furthermore, while the proposed method enables near-infrared (NIR) band prediction using RGB inputs, it should be noted that true NIR data remain necessary during the training phase. The key advantage of pseudo-NIR is that it can operate without dedicated NIR sensors once trained, offering a cost-effective and scalable solution for real-world deployment. However, pseudo-NIR should be considered an approximation and may lack the physical precision and interpretability of true NIR, particularly under unfamiliar environmental conditions. Future work should also investigate how pseudo-NIR outputs correlate with real-world vegetation indicators, such as growth stages or responses to rainfall, to better assess their practical utility.

## Figures and Tables

**Figure 1 sensors-25-04076-f001:**
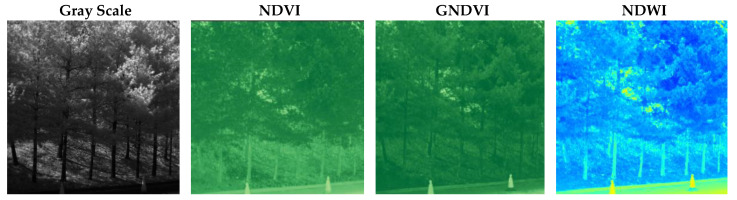
Comparison of grayscale, NDVI, GNDVI, and NDWI representations.

**Figure 2 sensors-25-04076-f002:**
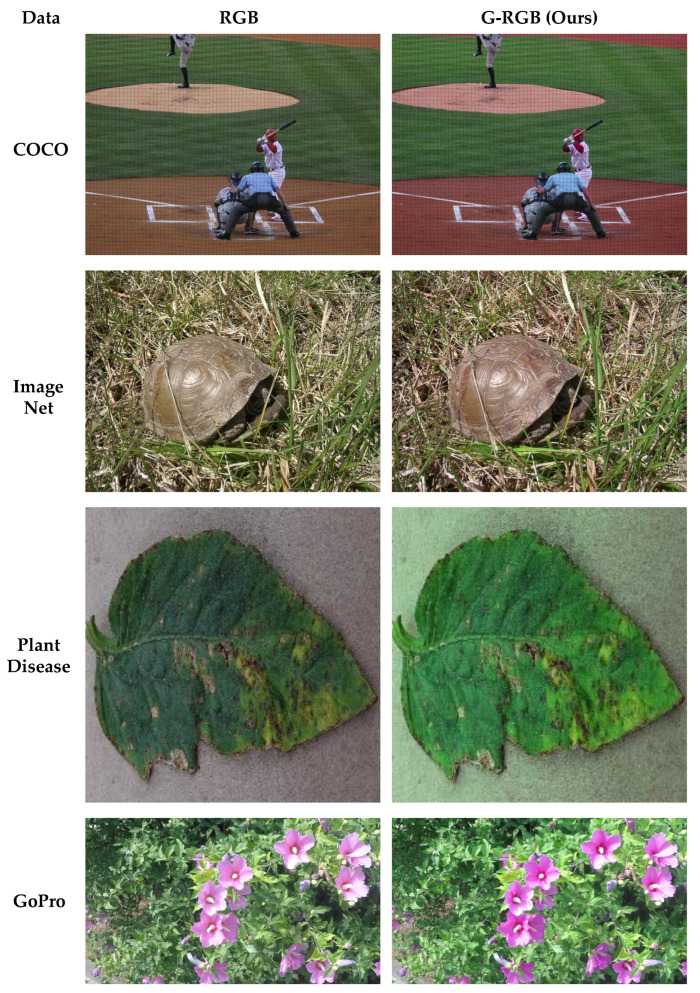
Comparison of RGB and G-RGB transformed images.

**Figure 3 sensors-25-04076-f003:**
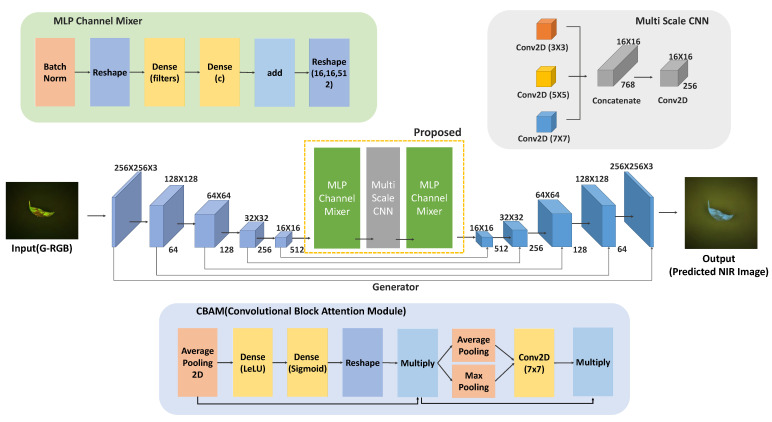
The proposed FUSE-Net architecture for NIR band prediction from G-RGB inputs, incorporating CBAM modules, MLP Channel Mixers, and a Multi-Scale CNN bottleneck for enhanced spectral–spatial feature learning.

**Figure 4 sensors-25-04076-f004:**
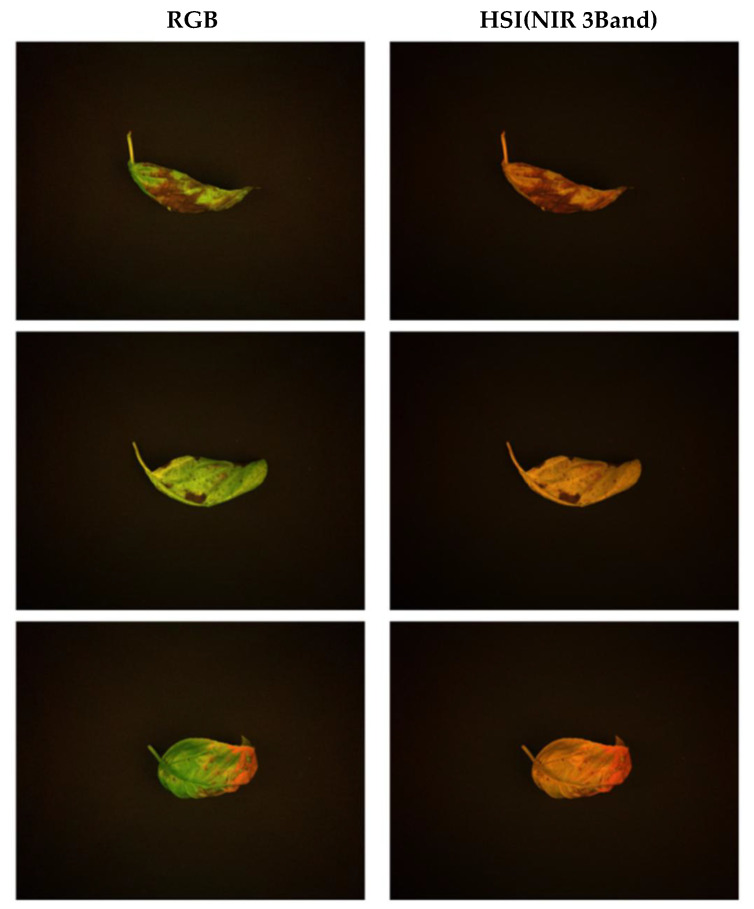
Representative samples of RGB and HSI images used in the experiment.

**Figure 5 sensors-25-04076-f005:**
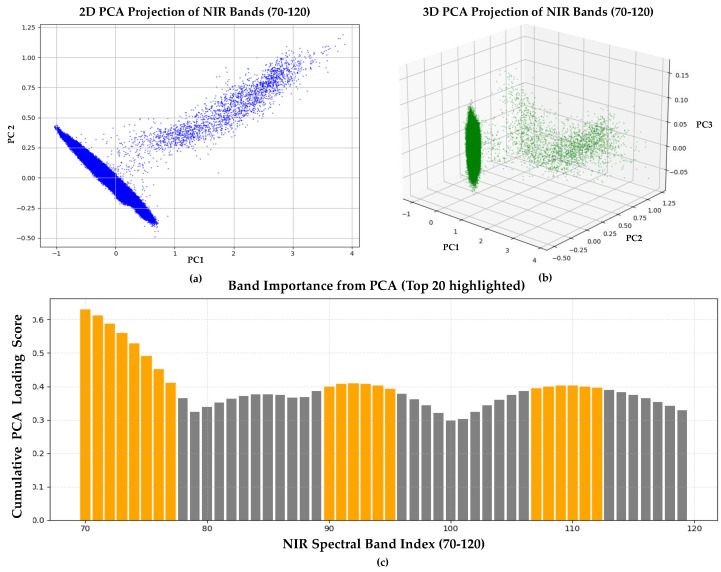
(**a**) Two-dimensional PCA projection of NIR spectral bands (70–120) onto the first two principal components (PC1 and PC2). (**b**) Three-dimensional PCA projection of NIR spectral bands (70–120) onto the first three principal components (PC1, PC2, and PC3). (**c**) Cumulative PCA loading scores for each spectral band (70–120). The top 20 bands with the highest scores, indicating the most informative spectral bands, are highlighted in orange.

**Figure 6 sensors-25-04076-f006:**
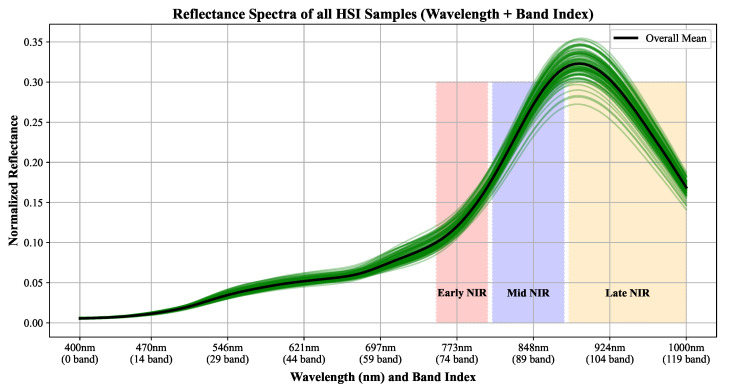
Average reflectance spectra of all HSI samples across the full wavelength range (400–1000 nm). The box regions represent the three NIR subgroups defined in this study: Early NIR (70–80), Mid NIR (81–95), and Late NIR (96–120). This division guided the construction of NIR band combinations for ablation testing.

**Figure 7 sensors-25-04076-f007:**
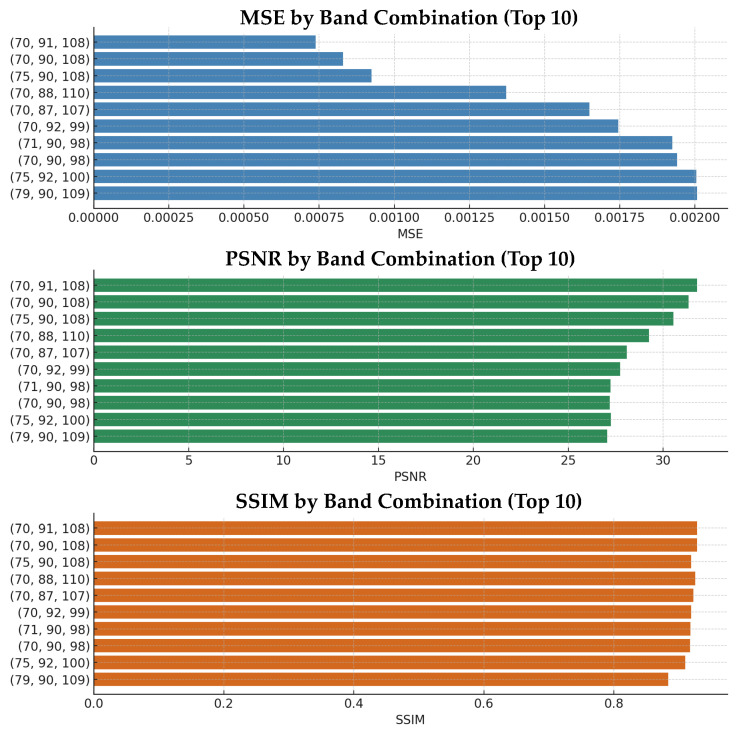
Comparison of 10 selected NIR band combinations using MSE, PSNR, and SSIM metrics.

**Figure 8 sensors-25-04076-f008:**
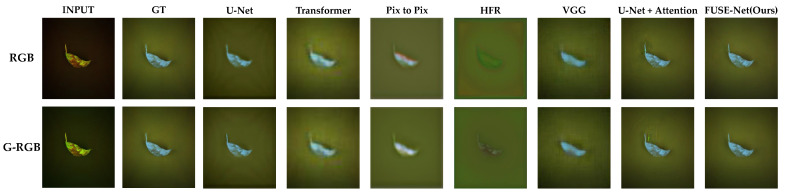
Visual comparison of NIR predictions using RGB and G-RGB inputs across different models.

**Figure 9 sensors-25-04076-f009:**
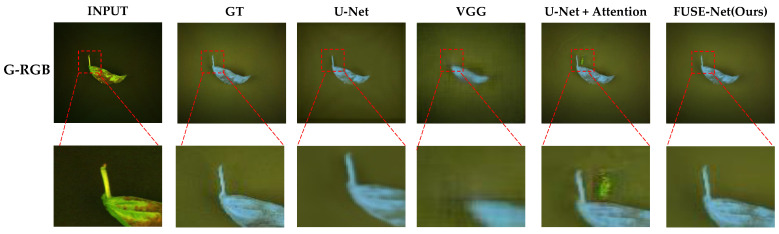
Close-up view of NIR predictions from top-performing models, highlighting structural details.

**Figure 10 sensors-25-04076-f010:**
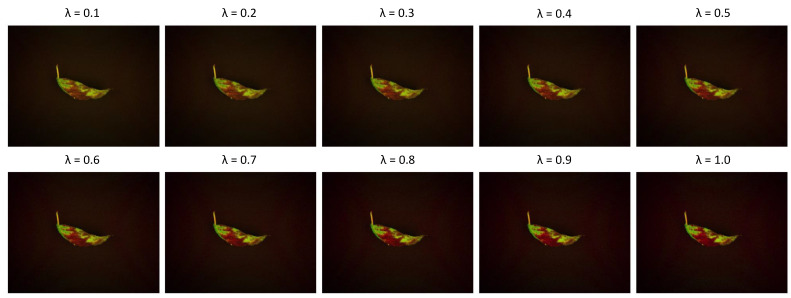
Qualitative comparison of G-RGB images generated with different λ values (0.1 to 1.0).

**Figure 11 sensors-25-04076-f011:**
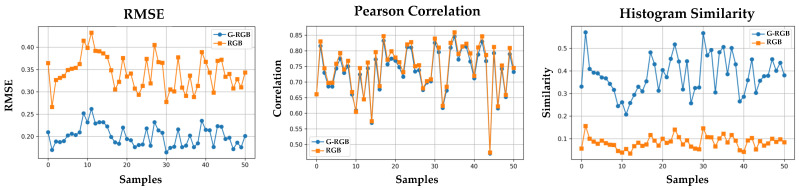
Sample-wise comparison of GNDVI similarity metrics between RGB-based and G-RGB-based NIR predictions. The figure shows (**top**) RMSE, (**middle**) Pearson correlation, and (**bottom**) histogram similarity, each computed against the ground truth GNDVI derived from true NIR.

**Table 1 sensors-25-04076-t001:** Quantitative evaluation results for the effectiveness of the proposed G-RGB method.

Dataset	L1 (MAE)	L2 (RMSE)	Histogram Similarity	Histogram Similarity (G)
COCO	4.1304	7.8613	0.9273	0.7820
ImageNet	2.3101	4.7969	0.9936	0.9809
Plant Diseases	2.5291	5.0411	0.9747	0.9241
GoPro	5.1010	10.3144	0.9868	0.9603

**Table 2 sensors-25-04076-t002:** Training hyperparameters.

Parameter	Value
Input Size	Transformer: (64, 64, 3)
	Pix2Pix: (128, 128, 3)
	Others: (256, 256, 3)
Output Channels	3 (Target NIR bands)
Optimizer	Adam
Data Augmentation	RandomFlip, Rotation, Zoom (TensorFlow built-in preprocessing)
Initial Learning Rate	0.0002
Loss Function	MSE (Mean Squared Error)
Evaluation Metrics	MAE, PSNR, SCC, SSIM
Batch Size	16
Epochs	Up to 450
EarlyStopping	patience = 30, monitored on val_loss, restore best weights
ReduceLROnPlateau	monitor = val_loss, factor = 0.5, patience = 10
Validation Split	0.2 (80% training, 20% validation)

**Table 3 sensors-25-04076-t003:** Quantitative evaluation of the top six NIR band combinations based on MSE, PSNR, and SSIM metrics.

Band Index	MSE	PSNR	SSIM
**(70, 91, 108)**	**0.0007**	**31.79**	**0.9281**
(70, 90, 108)	0.0008	31.33	0.9276
(75, 90, 108)	0.0009	30.54	0.9188
(70, 88, 110)	0.0013	29.25	0.9252
(70, 87, 107)	0.0016	28.07	0.9275
(70, 92, 99)	0.0017	27.72	0.9187

Bold values indicate the best performance.

**Table 4 sensors-25-04076-t004:** Performance comparison results by model based on Original RGB and G-RGB input.

Model	Input	MSE	PSNR	SCC	SSIM
U-Net	RGB	0.0011	29.85	0.9439	0.9264
	G-RGB	0.0011	29.64	0.9493	0.9120
Transformer	RGB	0.0013	29.12	0.9767	0.9247
	G-RGB	0.0012	29.40	0.9834	0.9225
Pix2Pix	RGB	0.0113	19.52	0.7257	0.8078
	G-RGB	0.0120	19.35	0.7190	0.8279
HFR	RGB	0.0110	19.64	0.4371	0.8432
	G-RGB	0.0099	20.09	0.4558	0.8652
VGG	RGB	0.0008	30.92	0.9442	0.8968
	G-RGB	0.0008	31.09	0.9464	0.9079
U-Net + Attention	RGB	0.0007	31.30	0.9835	0.9475
	G-RGB	0.0004	34.11	0.9910	0.9564
FUSE-Net (Ours)	RGB	0.0003	34.73	0.9887	0.9406
	**G-RGB**	**0.0001**	**37.89**	**0.9940**	**0.9605**

Bold values indicate the best performance.

**Table 5 sensors-25-04076-t005:** K-Fold performance results of each model based on original RGB and G-RGB inputs.

Model	Input	MSE	PSNR	SCC	SSIM
U-Net	RGB	0.0008±0.0003	31.31±0.71	0.9279±0.0163	0.9416±0.0029
	G-RGB	0.0008±0.0002	31.19±0.70	0.9286±0.0122	0.9448±0.0031
Transformer	RGB	0.0174±0.0054	17.87±1.21	0.1947±0.3460	0.6451±0.0677
	G-RGB	0.0095±0.0050	21.54±4.70	0.2732±0.4865	0.7636±0.0980
Pix2Pix	RGB	0.0039±0.0024	24.92±2.61	0.8127±0.0095	0.9103±0.0237
	G-RGB	0.0038±0.0021	25.02±2.79	0.6957±0.0515	0.8610±0.0242
HFR	RGB	0.0053±0.0045	25.45±6.41	0.3356±0.1509	0.8760±0.0523
	G-RGB	0.0038±0.0041	27.73±6.26	0.1685±0.3708	0.9139±0.0340
VGG	RGB	0.0014±0.0005	28.89±0.61	0.9522±0.0095	0.9071±0.0070
	G-RGB	0.0013±0.0004	29.22±0.52	0.9547±0.0103	0.9107±0.0080
U-Net + Attention	RGB	0.0027±0.0053	32.51±7.48	0.9828±0.0003	0.9380±0.0556
	G-RGB	0.0004±0.0004	35.12±0.38	0.9286±0.1132	0.9614±0.0014
FUSE-Net (Ours)	RGB	0.0021±0.0040	32.68±6.94	0.9826±0.0002	0.9445±0.0416
	**G-RGB**	0.0003±0.0004	36.02±0.73	0.9829±0.0041	0.9635±0.0011

Bold values indicate the best performance.

**Table 6 sensors-25-04076-t006:** Performance of FUSE-Net variants with different architectural components using RGB input. ✓ indicates the module is present.

Model Variant	MLP Mixer	Multi-Scale CNN	MSE ↓	PSNR ↑	SCC ↑	SSIM ↑
FUSE-Net (Variant A)	×	×	0.0029	25.42	0.9752	0.9313
FUSE-Net (Variant B)	×	✓	0.0026	25.78	0.9821	0.9415
FUSE-Net (Variant C)	✓	×	0.0024	25.97	0.9833	**0.9441**
**FUSE-Net (Full)**	✓	✓	**0.0003**	**34.73**	**0.9887**	0.9406

Bold values indicate the best performance.

**Table 7 sensors-25-04076-t007:** Sensitivity analysis of the λ parameter in the G-RGB transformation. The model was evaluated for different λ values using MSE, PSNR, and SSIM. Best performance is in bold.

λ in Equation (Equation 3)	MSE ↓	PSNR ↑	SSIM ↑
0.1	0.0162	19.88	0.811
0.2	0.0121	20.73	0.828
0.3	0.0098	21.55	0.839
0.4	0.0082	22.16	0.846
**0.5**	**0.0075**	**22.55**	**0.850**
0.6	0.0078	22.38	0.847
0.7	0.0087	22.01	0.842
0.8	0.0099	21.50	0.834
0.9	0.0117	20.84	0.825
1.0	0.0138	20.27	0.812

Bold values indicate the best performance.

**Table 8 sensors-25-04076-t008:** Average similarity between predicted and ground-truth GNDVI maps (true NIR), across 50 test samples.

Input-to-GNDVI Mapping	RMSE to GT ↓	Pearson Corr. to GT ↑	Histogram Sim. to GT ↑
RGB → Pseudo-NIR → GNDVI	0.343	**0.745**	0.084
G-RGB → Pseudo-NIR → GNDVI	**0.201**	0.733	**0.381**

Bold values indicate the best performance.

## Data Availability

The raw data supporting the conclusions of this article will be made available by the authors on request.

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
