# Peer review of "FUSE-Net: Multi-Scale CNN for NIR Band Prediction from RGB Using GNDVI-Guided Green Channel Enhancement"

_sensors, 2025, doi:10.3390/s25134076_

Round 1
Reviewer 1 Report
Comments and Suggestions for Authors
This paper proposes a Green Normalized Difference Vegetation Index (GNDVI)-guided green channel adjustment method, called G-RGB, to enable modiffes the spectral characteristics of the green channel to emphasize NIR-like information, thereby generating an enhanced input representation. Based on this, design a novel deep learning model, FUSE-Net, which integrates multi-scale convolutional layers and MLP-Mixer-based channel learning to capture spatial and spectral correlations effectively. This model’s performance is significantly improved and achieves promising results. Overall, the article is well written. But some technical details are not well described. Including but not limited to the following problems:
- Page 5 of 20, line 190,Why are there three [0,1]?
- The CBAM module is not indicated in Fig.3
- For each group, one band was randomly selected to form a three-band combination,it has 11×15×25 groups,Why choose only twenty groups? Is there any possibility of randomness?
- The additional 5-fold cross-validation should be described clearly.
- The datasets used to verify the feasibility of the model is not comprehensive enough; Are there other datasets to verify beyond it? This generalization is not obvious enough.
- Page 3 of 20, line107 the detail of NDWI should be added.
- PCA should be described clearly.
- The CBAM module in Figure 3 is not elaborated in detail.
- The theoretical basis for approximating NIR ()) through linear combination in Formula (2) is not fully explained. Why is the difference between Green and Red chosen instead of other bands?
- The dimension changes of the MLP layer and the role of residual connections in Formula (5) are not described in detail.
- The methods compared in the contrast experiments are not novel enough. It is suggested to compare with models that have performed well in recent years.
- The current experiments are only verified on a custom dataset and the cross-domain generalization ability has not been tested.
- Only MSE, PSNR, and SSIM are used, lacking specific indicators in this research field.
- Only 50 pairs of RGB-HSI images are used in the experiments, which may affect the model's generalization ability.
- The shortcomings of NDVI, GNDVI and NDWI in the text should be described in detail.
- The core difference between G-RGB and the existing RGB-NIR conversion methods is not clear.
- The advantages of MLP-Mixer in spectral reconstruction lack theoretical analysis.
- Add a sensitivity analysis of the impact of λ value on performance.
- It is suggested to add ablation experiments to clarify the respective contributions.
- The λ value in formula (3) is based on experimental verification, but the specific optimization method is not explained.
- Why was U-net chosen as the backbone.
Reviewer 2 Report
Comments and Suggestions for Authors
The main contribution by the authors seems to be the generation of NIR-like images from RGB inputs.
Of course, it is possible to compute a pseudo NIR reflectance. However, the question arises what are the verses and vices of this technique, when the true NIR information is simply unavailable. The authors did not find an answer to this question so far.
It is necessary to make quantitative comparison of the proposed FUSE technique with images where the NIR information is available. For example, how much better one can classify the cumulative effective of rainfall on vegetation using true NIR information and the proposed pseudo NIR from G-RGB images?
The authors implicitly conclude in the first lines of the abstract that the computation of the green Normalized Difference Vegetation Index counter measures the high costs for hyperspectral imaging equipment. This statement is very limited merit without proof.
line 173: Why is lambda fixed to 0.5? What happens if one choose a different kambda. To be investigated.
Above issues are to be tackled.
Round 2
Reviewer 2 Report
Comments and Suggestions for Authors
The authors tackled most of the open issue but one major:
"2. FUSE method should be quantitatively compared with true NIR (e.g., cumulative rainfall effect
on vegetation)"
The authors' response is not reflected in the body of the manuscript, as the authors replied. However, a conjecture without comparison has very little merit. This issue is still to be tackled in the manuscript.
Author Response
We sincerely thank the reviewer for the valuable follow-up comments. In the revised version, we have highlighted the main changes in red for ease of review.
Comment:
"2. FUSE method should be quantitatively compared with true NIR (e.g., cumulative rainfall effect on vegetation)"
The authors' response is not reflected in the body of the manuscript, as the authors replied. However, a conjecture without comparison has very little merit. This issue is still to be tackled in the manuscript.
Response:
As suggested, we have now included the relevant experiment results in the manuscript (Section 6.5.3 and Section 7, pages 20–21 and 21, lines 589–607 and 621–623). This section presents a quantitative comparison of the predicted NIR bands using GNDVI maps derived from both RGB and G-RGB inputs, evaluated against ground truth NIR-based GNDVI. We also briefly mention this experiment in the discussion section. We invite the reviewer to kindly examine these additions and would be grateful for any further feedback or suggestions.
Round 3
Reviewer 2 Report
Comments and Suggestions for Authors
the authors tackled all open issues